# Extending carbon chemistry at high-pressure by synthesis of CaC$_2$ and Ca$_3$C$_7$ with deprotonated polyacene- and *para*-poly(indenoindene)-like nanoribbons

Saiana Khandarkhaeva[1,2], Timofey Fedotenko[3], Alena Aslandukova [1], Fariia Iasmin Akbar [1], Maxim Bykov[4], Dominique Laniel[2,5], Andrey Aslandukov [2], Uwe Ruschewitz [4], Christian Tobeck[4], Björn Winkler[6], Stella Chariton[7], Vitali Prakapenka [7], Konstantin Glazyrin [3], Carlotta Giacobbe[8], Eleanor Lawrence Bright[8], Maxim Belov[9], Natalia Dubrovinskaia [2,9] & Leonid Dubrovinsky [1] ✉

Metal carbides are known to contain small carbon units similar to those found in the molecules of methane, acetylene, and allene. However, for numerous binary systems ab initio calculations predict the formation of unusual metal carbides with exotic polycarbon units, [C$_6$] rings, and graphitic carbon sheets at high pressure (HP). Here we report the synthesis and structural characterization of a HP-CaC$_2$ polymorph and a Ca$_3$C$_7$ compound featuring deprotonated polyacene-like and *para*-poly(indenoindene)-like nanoribbons, respectively. We also demonstrate that carbides with infinite chains of fused [C$_6$] rings can exist even at conditions of deep planetary interiors ( ~ 140 GPa and ~3300 K). Hydrolysis of high-pressure carbides may provide a possible abiotic route to polycyclic aromatic hydrocarbons in Universe.

Organic chemistry knows more than 10 million compounds and their number increases by 200–300 thousand annually. The richness of carbon chemistry and the existence of the vast number of organic compounds in nature are due to unlimited possibilities of carbon catenation resulting in the formation of molecules, clusters, chains, and rings differing in their length and complexity. Very frequently, simple structural motifs of these carbon units, e.g., C$_6$ rings, are repeating in organic compounds. High pressures (those above 100,000 atmospheres, ~10 GPa) have never been considered to promote the formation of new classes of organic materials. There are,

however, indications that high pressure can alter the bonding patterns in carbides, leading to new compounds with unusual structural units and interesting properties.

Carbides of the alkali and alkaline earth metals with a mainly ionic bonding character contain the simplest possible carbon anions[1–6]: isolated carbon atoms in methanides with C$^{4-}$ anions (in Be$_2$C or Al$_4$C$_3$), [C$_2$]$^{2-}$-dumbbells in acetylides (e.g., in CaC$_2$, SrC$_2$, BaC$_2$, Li$_2$C$_2$, Na$_2$C$_2$), or linear trimers [C$_3$]$^{4-}$ isoelectronic to CO$_2$ in allenides (with Mg$_2$C$_3$ being the only well-documented example)[7]. In rare earth metal carbides with metallic properties similar carbon units are found: C$_2$

[1]Bayerisches Geoinstitut, University of Bayreuth, Universitätstraße 30, 95440 Bayreuth, Germany. [2]Material Physics and Technology at Extreme Conditions, Laboratory of Crystallography University of Bayreuth, Universitätstraße 30, 95440 Bayreuth, Germany. [3]Deutsches Elektronen-Synchrotron DESY, Notkestraße. 85, 22607 Hamburg, Germany. [4]Institute of Inorganic Chemistry, University of Cologne, Greinstraße 6, 50939 Cologne, Germany. [5]Centre for Science at Extreme Conditions and School of Physics and Astronomy, University of Edinburgh, Edinburgh, UK. [6]Institute of Geosciences, Goethe University Frankfurt, Altenhöferallee 1, 60438 Frankfurt, Germany. [7]Center for Advanced Radiation Sources, The University of Chicago, 5640 S. Ellis, 60637 Chicago, IL, USA. [8]European Synchrotron Radiation Facility, CS 40220, 38043, Grenoble Cedex 9, France. [9]Department of Physics, Chemistry and Biology (IFM), Linköping University, SE-581 83 Linköping, Sweden. ✉e-mail: leonid.dubrovinsky@uni-bayreuth.de

dumbbells (e.g., in YC$_2$, LaC$_2$, La$_2$C$_3$ ≡ La$_4$(C$_2$)$_3$, CeC$_2$, TbC$_2$, YbC$_2$, and LuC$_2$)[2–4], isolated carbon atoms and (almost) linear C$_3$ units (in Me$_4$C$_7$ with Me = Y, Ho, Er, Tm, Lu)[5,6], or, most remarkably, C$_1$–C$_3$ units in Sc$_3$C$_4$ and Ln$_3$C$_4$ (Ln = Ho–Lu)[8]. However, due to the metallic bonding character, the assignment of ionic charges is not as straightforward as for the salt-like carbides mentioned above. At high pressure, for the binary systems Mg–C[9], Ca–C[10], Y–C[11], La–C[12], and Li–C[13] ab initio structure search predicts the formation of unusual metal carbides with exotic [C$_4$] and [C$_5$] units, [C$_6$] rings, and graphitic carbon sheets, as well as a number of structural transitions so that studying them under compression might enable to explore novel catenations of carbon atoms.

Calcium carbide has been the focus of high-pressure research during recent years. CaC$_2$ is of particular interest, as it is characterized by a rich polymorphism. Three polymorphs of CaC$_2$ (CaC$_2$-I, CaC$_2$-II, CaC$_2$-III) are known at ambient conditions[14]. Their structures feature [C$_2$]$^{2-}$ dumbbells common for acetylides. It was proposed that at relatively low pressures, these [C$_2$]$^{2-}$ units start to polymerize, and at about 7 GPa, CaC$_2$ transforms into a metallic phase (space group *Cmcm*) with 1D zig–zag polymeric carbon chains, as established using Raman spectroscopy and X-ray diffraction (XRD)[15]. (Note that in ref. 15 at ~10 GPa, the signal of the C≡C stretching vibration is still dominating, and the quality of the resulting diffraction patterns is too low for an unambiguous Rietveld fit). Theory predicts further structural transformations in CaC$_2$ upon pressure increase coupled with the progressive polymerization of carbon atoms. According to calculations by Li et al.[16], above 20 GPa, the structure of CaC$_2$ should adopt the *P*$\bar{1}$ symmetry and contain infinite carbon strips built of interconnected pairs of fused five-membered rings. This phase was supposed to be thermodynamically stable up to 37 GPa[16]. Above this pressure, according to the calculation, metallic *P*$\bar{1}$-CaC$_2$[13,16] should transform into metallic *Immm*-CaC$_2$[13,16], in which carbon atoms are polymerized to form infinite quasi-1D ribbons built of fused six-membered rings. The latter prediction is of particular interest, as the suggested carbon catenation in *Immm*-CaC$_2$[13,16] resembles deprotonated polyacene-like nanoribbons.

Here, we report the results of high-pressure high-temperature (HPHT) studies of CaC$_2$ in laser-heated diamond anvil cells (LHDACs) performed using synchrotron single-crystal X-ray diffraction (SCXRD) up to ~150 GPa at temperatures up to ~3000 K. The predicted *Immm*-CaC$_2$[13,16] high-pressure polymorph of CaC$_2$ (HP-CaC$_2$) containing deprotonated polyacene nanoribbons was synthesized and fully characterized. A novel Ca$_3$C$_7$ compound, never anticipated or observed before, with an orthorhombic crystal structure (space group *Pnma*) featuring infinite, fully deprotonated *para*-poly-indenoindene (*p-PInIn*)-like chains was discovered, and its structure was solved and refined using SCXRD. We demonstrate that compounds with fused six-membered (reminiscent in shape of benzene) carbon rings can be synthesized above 100 GPa. We demonstrated that on decompression in the presence of water, HP-CaC$_2$ forms polycyclic aromatic hydrocarbons (PAHs).

## Results and discussion

Several diamond anvil cells with different samples of calcium carbide were prepared as described in "Methods". Studies cover the pressure range up to ~150 GPa. A summary of all experiments presented in this work, including pressure–temperature (*P–T*) conditions of the synthesis and the unit cell parameters of the observed phases, is given in Supplementary Table S1.

Upon compression to the target pressures, we performed Raman spectroscopy measurements of the starting material (Supplementary Fig. 1). In agreement with previous reports[14,16] at pressures below ~5 GPa we observed spectral signatures of co-existing tetragonal CaC$_2$-I and monoclinic CaC$_2$-II phases; above ~10 GPa only CaC$_2$-I was observed, and at ~25 GPa the Raman spectrum became featureless

(suggesting amorphization of CaC$_2$ as proposed in ref. 14). Microphotographs taken during compression reveal changes in the sample's visual appearance (Supplementary Fig. 1): the crystal, which was initially transparent (3.3 GPa), became translucent at ~10 GPa and then opaque above ~25 GPa. Such changes can be attributed to the progressive band gap closure in CaC$_2$ upon compression that agrees with resistivity measurements performed by Zheng et al.[17].

The systematic analysis of the XRD data obtained from different DACs after laser heating at different pressures (Supplementary Table 1) allowed identifying diamond (as a product of paraffin oil decomposition), CaO (in some cells, due to possible contamination of the commercial calcium carbide with the oxide or Ca(OH)$_2$), and two new phases: a high-pressure polymorph of CaC$_2$ (called here HP-CaC$_2$) and a new Ca–C compound Ca$_3$C$_7$. After laser heating, the sample remains dark and non-transparent (Supplementary Fig. 9). The results of different experiments are mutually consistent and reproducible. The structures of the new solids were solved and refined using the best quality SCXRD datasets, obtained at 44(1) GPa and 147(2) GPa for HP-CaC$_2$ and at 38(1) GPa for Ca$_3$C$_7$ (see Supplementary Tables 2, 3, and 5). They are described in detail below.

The structure of the novel HP-CaC$_2$ polymorph has an orthorhombic unit cell (space group *Immm*, Z = 4) with the lattice parameters *a* = 2.5616(3) Å, *b* = 6.0245(17) Å, *c* = 6.6851(15) Å at 44(1) GPa (Fig. 1a, Supplementary Fig. 2, and Supplementary Table 2). There are four calcium atoms in the unit cell (Fig. 1a), which occupy the Wyckoff site 4 *i* (0, 0, 0.2039(2)). The Ca–C distances to the ten nearest carbon atoms vary from ~2.44 Å to ~2.46 Å at 44(1) GPa. Carbon atoms occupy two distinct crystallographic positions (Fig. 1a, b): C1 – 4 *g* (0, 0.3824(14), 0) and C2 – 4 *h* (0, 0.2422(16), 0.5). They all are polymerized, forming infinite nanoribbons of fused planar six-membered rings (Fig. 1b). These planar ribbons, lying in the *ab* plane, are oriented along the *a*-axis. Thus, the structure can be described as an alternation of layers containing carbon nanoribbons and layers of calcium atoms along the *c*-axis. The described arrangement of carbon atoms gives rise to an exotic form unprecedented in any known carbide poly-C entity: a [C$_4$] unit with a formal charge of 4− (Supplementary Fig. 4). The C–C distances in the hexagons, refined at 44(1) GPa, are equal to ~1.41 Å and ~1.48 Å for the C1–C1 and C1–C2 bonds, respectively. Two angles in the hexagon are equal at ~120.5°, while the other four angles are ~119.8° (Fig. 1b). It has been shown that upon compression, the C$_6$-ring symmetrizes: the two bond lengths, which were different at 44(1) GPa, became identical and equal with ~1.41 Å at 144(2) GPa (Supplementary Table 5).

Laser-heating of CaC$_2$ in paraffin oil at pressures of ~38 GPa results in the appearance of new diffraction peaks, which belong neither to HP-CaC$_2$ nor to B1-CaO. Indexing reveals an orthorhombic unit cell with the lattice parameters *a* = 4.762(4) Å, *b* = 8.3411(11) Å, and *c* = 8.8625(13) Å at 38(1) GPa (space group *Pnma*) (see Supplementary Fig. 3, Supplementary Table 3). The structure solution and refinement evidence a Ca$_3$C$_7$ stoichiometry with four formula units per unit cell. There are two independent crystallographic positions of calcium atoms in the crystal structure of Ca$_3$C$_7$: Ca1 on the Wyckoff site 4 *c* (0.0517(3), 0.25, 0.33976(5)) and Ca2 on the Wyckoff site 8 *d* (0.1956(2), 0.5744(4), 0.3533(4)). The carbon atom C1 is located on the Wyckoff position 4 *c* (0.1913(12), 0.25, 0.6330(2)). The other carbon atoms C2 (0.2158(8), 0.0535(2), 0.1205(2)), C3 (0.1818(8), 0.6127(2), 0.06479(18)) and C4 (0.0172(8), 0.1627(2), 0.04884(18)) occupy the general Wyckoff sites 8d (*x,y,z*). In the crystal structure of Ca$_3$C$_7$, carbon atoms form infinite non-planar chains made of fused distorted C$_6$ and C$_5$ rings, which are isomorphous to deprotonated *para*-type poly(indenoindene) (*p-PInIn*)[18] (Fig. 2). The projection on the [011] plane visualizes the alternation of layers of calcium atoms with deprotonated *p-PInIn* carbon chains (Fig. 2). To the best of our knowledge, such type of polycarbide entity has never been predicted or reported for any of the metal carbides known in inorganic

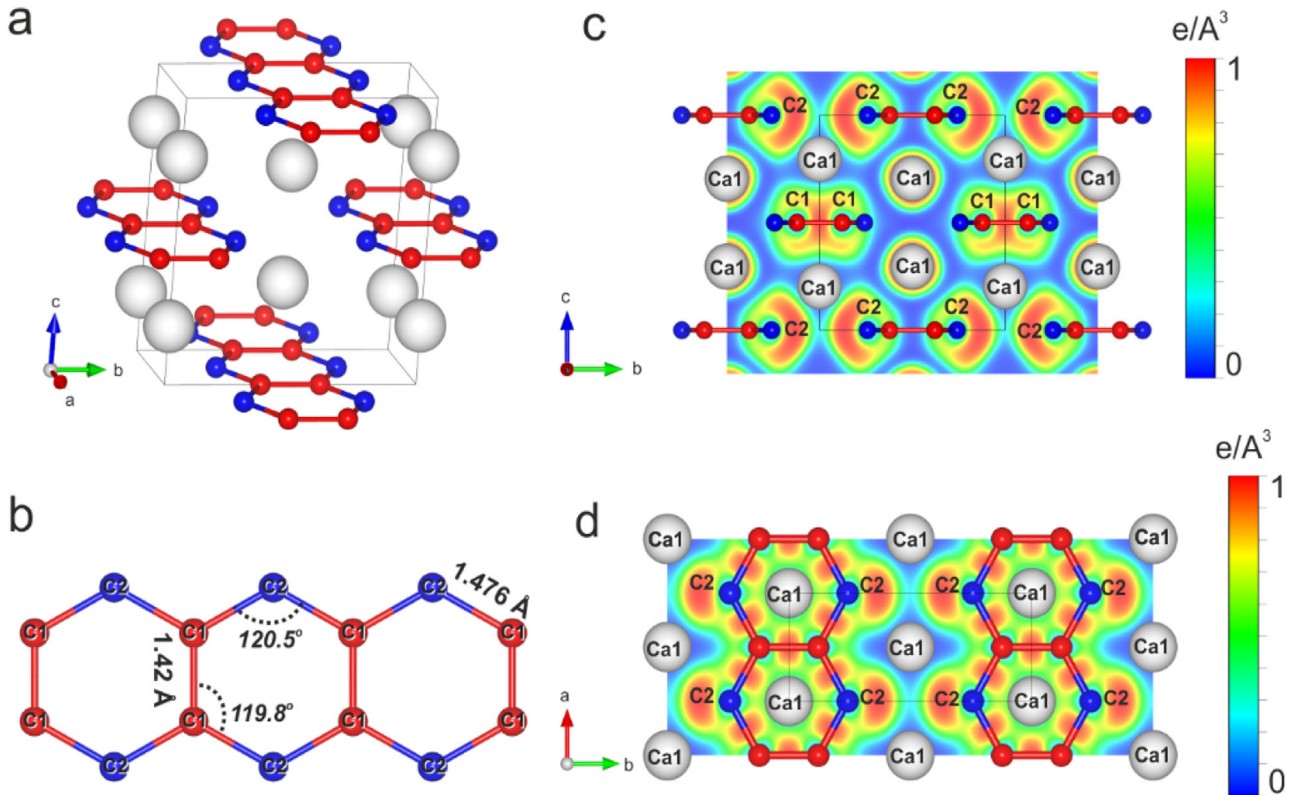

**Fig. 1 | Crystal structure of HP-CaC$_2$ at 44(1) GPa. a** A ball and stick model with the unit cell outlined; calcium atoms are shown as white spheres, and carbon atoms as red and blue balls for two distinct crystallographic positions, C1 (Wyckoff site 4 $g$) and C2 (4 $h$), respectively. **b** The geometry of a single deprotonated polyacene nanoribbon; the C–C distances and C-C-C angles are labeled. Cross-sections of the calculated electron localization function (ELF) are shown in the planes perpendicular (**c**) and parallel (**d**) to the polyacene nanoribbons.

chemistry[19–21]. There are five different C–C distances in the complex [C$_{14}$] polyanion with the formal charge 12− (Supplementary Fig. 5); at 38(1) GPa, the shortest and the longest C–C bonds are equal to ~1.43 Å and ~1.48 Å, correspondingly. Unlike the C$_6$-rings in the deprotonated polyacene nanoribbons of HP-CaC$_2$, the carbon hexagon in the *p-PInIn* anion of Ca$_3$C$_7$ exhibits pronounced distortion, where two angles strongly deviate from 120° with ~111.6°. These modest differences underline a subtle but recognizable dissimilarity in their chemical environment, namely the complexity of poly-C species and their coordination by Ca$^{2+}$ ions.

Theoretical calculations perfectly reproduce experimental structural data (Supplementary Tables 2 and 3). The pressure dependences of the unit cell parameters and volumes of novel calcium carbides were derived from DFT calculations (Fig. 3). The results were described by the Birch–Murnaghan equation of state (EoS) with the parameters $V_0 = 127.17(2)$ Å$^3$, $K_0 = 144.5(2)$ GPa, and $K´ = 3.92(1)$ for HP-CaC$_2$ and $V_0 = 432.0(2)$ Å$^3$, $K_0 = 125(1)$ GPa, and $K´ = 4$ (fixed) for Ca$_3$C$_7$. As seen in Fig. 3, both compounds experience an anisotropic contraction upon compression, with the most incompressible directions being along the polyacene-like nanoribbons in HP-CaC$_2$ ($a$-axis) and deprotonated *p-PInIn* chains in Ca$_3$C$_7$ ($b$-axis). The unit cell parameters derived from several independent SCXRD experiments conducted at various pressures (Supplementary Table 1) show a good agreement with the calculated EoSes (Fig. 3).

Harmonic phonon dispersion calculations reveal that the novel calcium carbides are dynamically stable at their synthesis pressures (Fig. 4). The calculated electron density of states shows that HP-CaC$_2$ is metallic and the main contributions at the Fermi level come from the Ca $3d$ and C $2p$ states (Fig. 4). The computed charge density was analyzed in terms of the electron localization function (ELF), which revealed a strong covalent bonding between carbon atoms within deprotonated polyacene nanoribbons, and ionic bonds between Ca1 and C2 atoms (Fig. 1). Bader charge analysis at 40 GPa for Ca1, C1, and C2 atoms yields values of 1.191, −0.378, and −0.813 (Supplementary Table 4), respectively.

Calculated electron density of states of Ca$_3$C$_7$ suggests it to be a narrow-band $p$-type semiconductor (Fig. 4). Computed ELF charge density shows the formation of covalent bonds between carbon atoms in the deprotonated *p-PInIn* carbon chains, and strong ionic interaction between Ca and C atoms (Fig. 2). Bader charge analysis at 40 GPa for Ca1, Ca2 atoms yields values of 1.227, 1.232; for C1, C2, C3, C4 atoms: −0.718, −0.798, −0.338, −0.351, respectively (Supplementary Table 4).

To explore the thermodynamic stability of HP-CaC$_2$ and Ca$_3$C$_7$ in comparison to other calcium carbides, a convex hull was constructed for the binary Ca–C system at a pressure of ~40 GPa considering the predicted phases[16]. Both phases observed in the present study lie on the convex hull and therefore are (theoretically) thermodynamically stable at 40 GPa (Fig. 5). Notably, HP-CaC$_2$ also remains thermodynamically stable at 140 GPa (Supplementary Fig. 6). Although an attempt to recover HP-CaC$_2$ and Ca$_3$C$_7$ at ambient conditions was unsuccessful, according to our calculations, the compounds should be dynamically stable even at atmospheric pressure (Supplementary Fig. 7).

Polycyclic aromatic hydrocarbons (PAHs) consisting of a finite number of fused aromatic rings give rise to various classes of polymeric compounds, if a PAH is extended into a polymer. For example, a particular class of fully conjugated polymers is represented by an extension of indenofluorene into a polymer, that is poly(-indenoindene) (*PInIn*)[22], where the alternation of five- and six-membered rings gives rise to a ladder structure. Although its targeted synthesis in bulk has remained elusive so far, very recently, on-surface synthesis of *para*-type oligo(indenoindene) (*p-OInIn*) was

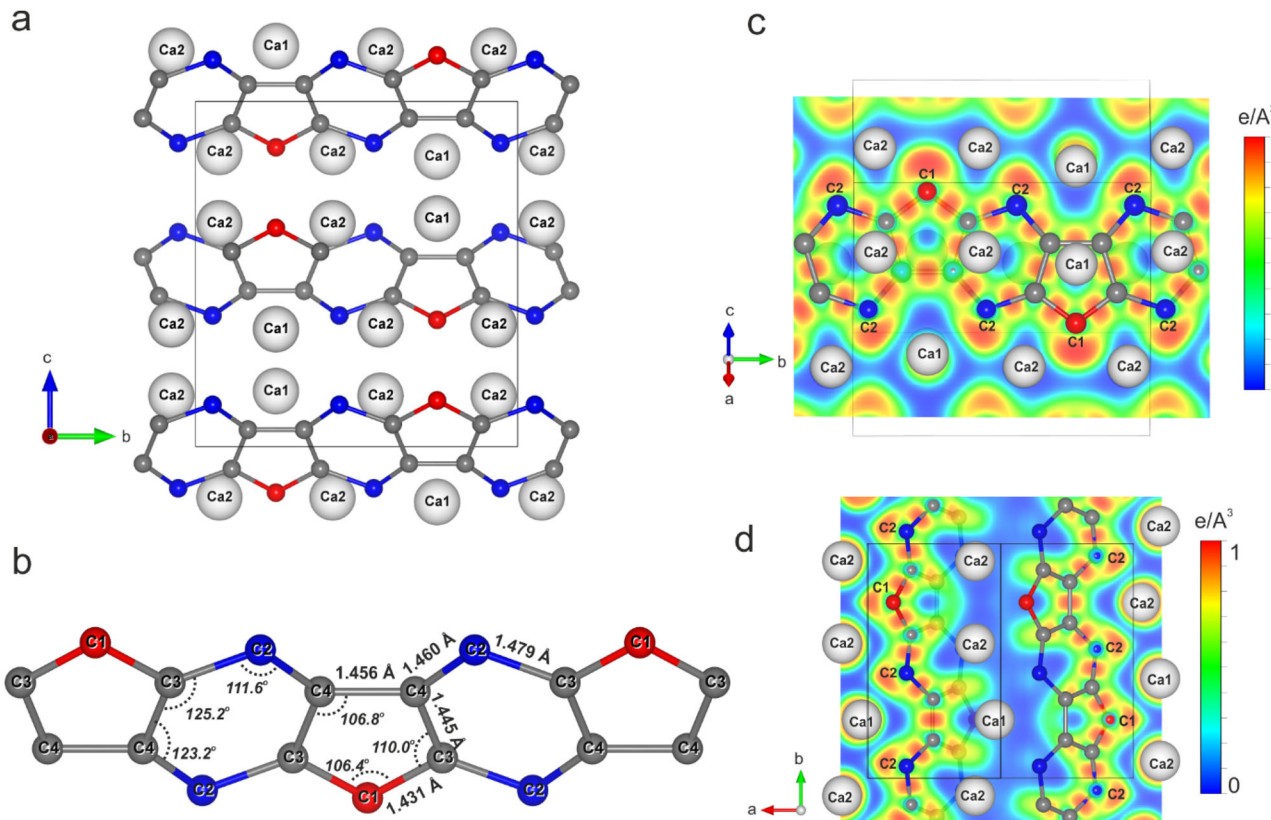

**Fig. 2 | Crystal structure of Ca₃C₇ at 38(1) GPa. a** A projection of the Ca₃C₇ structure along the *a*-axis, emphasizing 2D chains of carbon atoms aligned along the *b*-axis. Calcium atoms are shown as white spheres, and carbon atoms as red and blue balls for the two distinct crystallographic positions C1 (4*c*) and C2 (8*d*), respectively. Carbon atoms, named C3 (8*d*) and C4 (8*d*), are shown as gray balls.

**b** The geometry of a single deprotonated *para*-poly(indenoindene) (*p-PInIn*) chain with the C-C distances and C-C-C angles labeled. **c**, **d** Cross-sections of the calculated electron localization function (ELF) are shown in the two different planes containing *p-PInIn* chains.

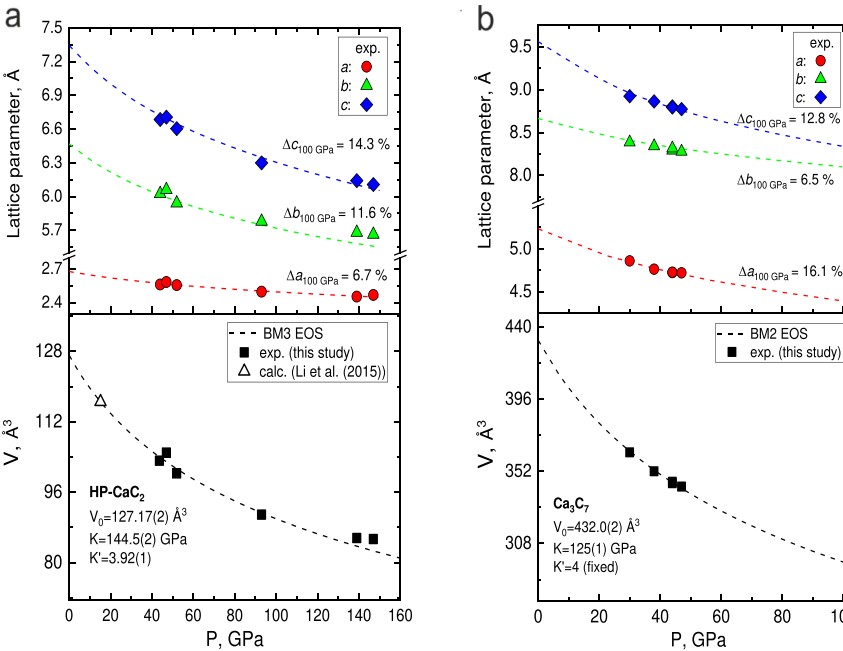

**Fig. 3 | Pressure dependence of the lattice parameters and the unit cell volumes for the two Ca−C compounds. a** HP-CaC₂; **b** Ca₃C₇. Solid symbols are experimental data points; dashed curves represent theoretical data; the black dashed lines are the fit of the calculated PV data using the third and second-order Birch−Murnaghan

equations of state with the following parameters: $V_0 = 127.17(2)$ Å³, $K_0 = 144.5(2)$ GPa, and $K´ = 3.92$ for HP-CaC₂; and $V_0 = 432.0(2)$ Å³, $K_0 = 125(1)$ GPa, $K´ = 4$ (fixed) for Ca₃C₇.

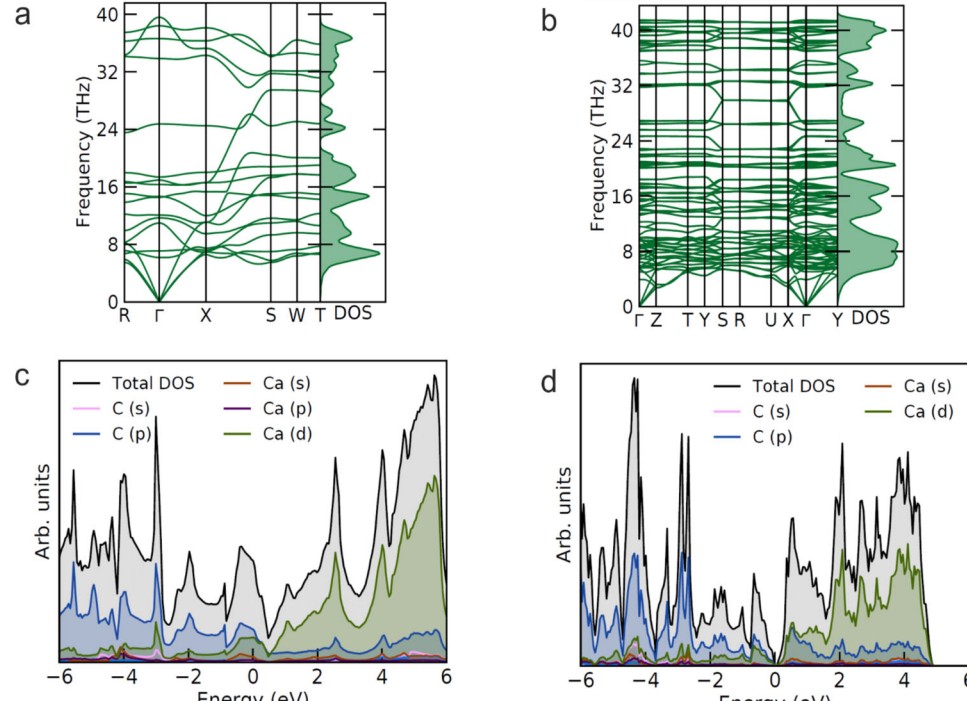

**Fig. 4 | Results of DFT-based calculations for HP-CaC$_2$ at 44(1) and Ca$_3$C$_7$ at 38(1) GPa.** Calculated phonon dispersion curves for HP-CaC$_2$ (**a**) and Ca$_3$C$_7$ (**b**). Calculated electron density of states for HP-CaC$_2$ (**c**) and Ca$_3$C$_7$ (**d**); the Fermi energy level was set at 0 eV.

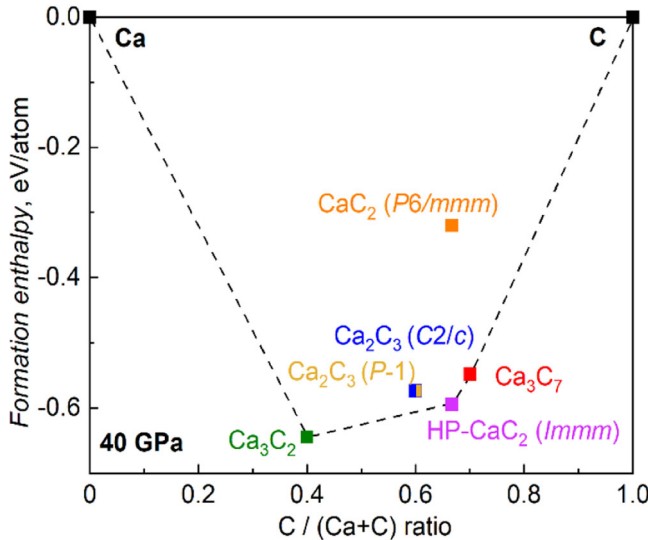

**Fig. 5 | The calculated convex hull for the Ca–C binary system with predicted and observed calcium carbides at 40 GPa.** The phases HP-CaC$_2$ and Ca$_3$C$_7$ lie on the convex hull (black dashed line) and are, thus, thermodynamically stable. The phases Ca$_2$C$_3$ (*P$\bar{1}$*), Ca$_2$C$_3$ (*C2/c*) and CaC$_2$ (*P6/mmm*) are off the convex hull and are, therefore, unstable.

reported to be successful through a sequence of thermally activated reactions from an organic precursor[18]. In the present work, we have demonstrated the inorganic synthesis of the Ca$_3$C$_7$ compound in bulk with the structure featuring deprotonated *para*-poly-indenoindene-like chains. The HP-CaC$_2$ synthesized in this work also gives a unique example of polyacene-like nanoribbon moieties previously unknown among any organic or metal-organic compounds. Moreover, the synthesis of single crystals of this material with fused carbon rings similar to those in benzene at pressures above 140 GPa demonstrates

that, eventually, organic chemistry may be extended to truly extreme conditions.

The fact that carbides (and eventually other classes of chemical compounds) with deprotonated polyacene- and *para*-poly(-indenoindene)-like nanoribbons can exist (and even form) at conditions of planets' interiors may have important implications for geosciences and astrobiology. Indeed, PAHs were detected at many locations in the Universe[23–26], but their origin (as well as a possible role in the emergence of life) remains controversial. The hydrolysis of carbides in planets' interiors, on their surfaces or atmospheres may be a source of hydrocarbons. Hydrolysis products of carbides as complex as HP-CaC$_2$ or Ca$_3$C$_7$ we discovered may contain PAHs. In order to test this hypothesis, a mixture of CaC$_2$-I and Ca(OH)$_2$ was loaded into a DAC chamber, compressed to ~40 GPa, and the whole sample was double-sided laser-heated between 2500 K and 3000 K. On decompression at ambient temperature, featureless Raman spectra were observed down to ~1.8 GPa (Supplementary Fig. 8). At this pressure, the DAC was warmed to 85 °C (above the melting point of water) using a hot plate for 72 h. Several of the Raman modes we observed (Supplementary Fig. 8) belong to C–H vibrations of saturated and aromatic hydrocarbons. Moreover, at some spots within the sample, the Raman spectra closely resemble those of pyrene (C$_{16}$H$_{10}$) (Supplementary Fig. 8), confirming a possible abiotic route to aromatic compounds through the hydrolysis of high-pressure carbides within planetary interiors.

## Methods
### Sample preparation
Calcium carbide (CaC$_2$) in the form of pieces of technical grade was purchased from Sigma-Aldrich. Single crystals with an average size of ~0.05 × 0.05 × 0.01 mm$^3$ were preselected using a three-circle Bruker diffractometer (SMART APEX CCD detector, Ag-Kα radiation, wavelength $\lambda$ = 0.5594 Å) installed in the Bayerisches Geoinstitut (BGI, Bayreuth, Germany). All manipulations with the crystals were conducted in paraffin oil to avoid their contamination by air

and/or moisture. Although XRD showed a phase purity of the selected samples (no other phases except $CaC_2$-I[13] were detected), we cannot exclude that they might contain some very small, undetectable amount of calcium oxide CaO or calcium hydroxide $Ca(OH)_2$. In order to exclude the existence of CaO and $Ca(OH)_2$ completely, we did additional experiments with a pre-synthesized $CaC_2$ sample of high purity. It was prepared from distilled calcium (70.6 mg, 1.76 mmol, 1 eq.) and a slight surplus of graphite (44.4 mg, 3.70 mmol, 2.1 eq.) heated at 800 °C in high vacuum for 24 h prior to the syntheses. Calcium and graphite were sealed in a purified tantalum ampoule (-0.7 bar argon pressure) and sealed under vacuum in a second ampoule made of quartz to avoid oxidation of the inner tantalum ampoule. These ampoules were heated at 1200 °C for 14 h. The purity of the products was checked by powder X-ray diffraction. It revealed that a mixture of $CaC_2$-I, $CaC_2$-II, and $CaC_2$-III had formed. No reflections indicating CaO or $Ca(OH)_2$ impurities were found. All handling of these air and moisture-sensitive samples was carried out in inert atmospheres (argon), and they were loaded without a pressure-transmitting medium. The preselected single crystals of commercial $CaC_2$ were loaded into BX90 diamond anvil cells[27] (DAC) equipped with pairs of Boehler–Almax diamond anvils with culet diameters of 250 μm or 120 μm. The gasket was made of chemically pure Re-foil indented to -25 μm thickness with a hole in the center of the indent of -110 or -60 μm in diameter. Paraffin oil was used as a pressure-transmitting medium. The in-house laser heating was performed in a continuous mode using a double-sided laser-heating setup at the BGI[28]. The NIR laser beam ($\lambda = 1070$ nm) was focused to a -5 μm spot, with the temperature determined according to the gray body approximation of Planck's law (Supplementary Fig. 10). As a rule, a sample easily absorbs laser light and its heating requires relatively low power (~5 to -15 W from each side, depending on pressure). Heating duration varies from -15 to -60 s. Two or three spots were heated at each sample, and transformed materials were found in all heated spots. The pressure was determined using the Raman signal from the diamond anvils[29] and additionally monitored by the equation of states (EOS) of Re gasket[30].

## Synchrotron X-ray diffraction studies
High-pressure SCXRD experiments were conducted on the Extreme Conditions Beamline P02.2 at PETRA III, Hamburg, Germany ($\lambda = 0.2885$ Å, beam size -1.5 × 1.7 μm² at FWHM, PerkinElmer XRD1621 detector), on the 13-IDD beamline at the Advanced Photon Source (APS), Chicago, USA ($\lambda = 0.2952$ Å, beam size -3 × 3 μm², Pilatus CdTe 1 M detector), and on the Material Science beamline ID11, ESRF, Grenoble, France ($\lambda = 0.2952$ Å, beam size -0.5 × 0.5 μm², Eiger 4 M CdTe detector). Prior to the collection of the SCXRD datasets, a 2D X-ray mapping was performed over the heated area in order to find the best spots for data acquisition. During the SCXRD measurements, the DACs were rotated about the vertical ω-axis in a range of ±38°, and the diffraction images were recorded with an angular step of $\Delta\omega = 0.5°$.

## XRD data processing
The SCXRD data analysis, including the indexing of diffraction peaks, data integration, frame scaling, and absorption correction, was performed using CrysAlisPro software package[31]. A crystal of orthoenstatite ($Mg_{1.93}Fe_{0.06}$)($Si_{1.93}Al_{0.06}$)$O_6$ (space group $Pbca$, $a = 8.8117(2)$ Å, $b = 5.18320(10)$ Å, $c = 18.2391(3)$ Å) was used as a calibration standard for refinement of the instrumental parameters of the diffractometer (the sample-to-detector distance, the detector's origin, offsets of the goniometer angles and rotations of the X-ray beam and the detector around the instrument axis). The structures of novel phases were solved with the SHELXT[32] structure solution program using intrinsic phasing and refined with the Jana2006 program[33]. Crystal structure visualization was made with the VESTA software[34].

## Raman spectroscopy
The spectra were collected using a LabRam system equipped with a He:Ne laser source (excitation wavelength of 632 nm) and a DilorXY system equipped with an Ar laser source (excitation wavelength of 514 nm). Laser power in the range of 5–15 mW with the He:Ne laser and up to 100 mW for the Ar laser were applied. Raman spectra were collected in the region 1100–2000 cm⁻¹ by means of 5 accumulations for 30 s each. The frequency resolution was 2 cm⁻¹.

## DFT calculations
The first-principles calculations were done using the framework of density functional theory (DFT) as implemented in the Vienna Ab initio Simulation Package (VASP)[35]. To expand the electronic wave function in plane waves, we used the Projector-Augmented-Wave (PAW) method[36]. The Generalized Gradient Approximation (GGA) function was used for calculating the exchange-correlation energies, as proposed by Perdew–Burke–Ernzerhof (PBE)[37]. The PAW potentials with the following valence configurations of 3s3p4s for Ca and 2s2p for C were used. Convergence tests with a threshold of 1 meV per atom in energy and 1 meV/Å per atom for forces led to a Monkhorst-Pack[38] $k$-point grid of 16 × 6 × 6 for $CaC_2$ and 9 × 5 × 5 for $Ca_3C_7$ and an energy cutoff for the plane wave expansion of 900 eV. The phonon frequencies and phonon band structure calculations were performed in the harmonic approximation with the help of PHONOPY software using the finite displacement method[39] for 3 × 3 × 3 ($CaC_2$) and 3 × 2 × 2 ($Ca_3C_7$) supercells with respectively adjusted $k$-points. For Brillouin zone integrations, meshes of 10 × 10 × 10 ($CaC_2$) and 10 × 6 × 6 ($Ca_3C_7$) $k$-points were used within the tetrahedron method. The charge densities obtained from VASP calculations were used for Bader's charge analysis in order to obtain the total charges associated with each atom and the zero flux surfaces defining the Bader volumes. Equation of state (EoS) and static enthalpy[40] calculations were performed via variable-cell structural relaxations between 0–100 GPa for $Ca_3C_7$ and 0–150 GPa for HP-$CaC_2$. In our calculations, temperature, configurational entropy, and the entropy contribution due to lattice vibrations were neglected.

## Data availability
Crystallographic data for the structures reported in this Article have been deposited at the Cambridge Crystallographic Data Center under deposition numbers CCDC 2282632, 2282633, and 2282634. Copies of the data can be obtained free of charge via https://www.ccdc.cam.ac.uk/structures/. Some structural data were reported in the PhD Thesis by Dr Saiana Khandarkhaeva https://epub.uni-bayreuth.de/id/eprint/7172/. All data used to plot figures can be obtained from the corresponding author via request.

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

## Acknowledgements

The authors acknowledge the Deutsches Elektronen-Synchrotron (DESY, PETRA III) for the provision of beamtime at the P02.2 beamline, the European Synchrotron Radiation Facility (ESRF) for the provision of beamtime at the ID11 beamline, and the Advanced Photon Source (APS) for the provision of beamtime at the 13ID-D beamline. Help in experiments at DESY by H.-P. Liermann is appreciated. Portions of this work were performed at GeoSoilEnviroCARS, Advanced Photon Source, and Argonne National Laboratory. GeoSoilEnviroCARS is supported by the National Science Foundation—Earth Sciences via SEES: Synchrotron Earth and Environmental Science (EAR –2223273). This research used resources from the Advanced Photon Source, a U.S. Department of Energy (DOE) Office of Science User Facility operated for the DOE Office of Science by Argonne National Laboratory under Contract No. DE-AC02-06CH11357. N.D., L.D., and U.R. thank the Deutsche Forschungsgemeinschaft (DFG) (DFG projects DU 954-11/1, DU 945/15-1; LA 4916/1-1; DU 393–9/2, ME 5206/3-1, RU 546/15-1) for financial support. N.D. also thanks the Swedish Government Strategic Research Area in Materials Science on Functional Materials at Linköping University (Faculty Grant SFO-Mat-LiU No. 2009 00971) for financial support. Partially funded by the Open Access Publishing Fund of the University of Bayreuth. MB acknowledges the support of the DFG Emmy-Noether project BY112/2-1. D.L. thanks the UKRI Future Leaders Fellowship (MR/V025724/1) for financial support. For the purpose of open access, the author has applied a Creative Commons Attribution (CC BY) license to any Author Accepted Manuscript version arising from this submission.

## Author contributions

S.K., L.D., and N.D. designed the research. U.R. and C.T. synthesized precursor material. S.K., T.F., and L.D. prepared the high-pressure experiments. S.K., T.F., An.A., Al.A., D.L., S.C., V.P., K.G., E.L.B., C.G., performed the synchrotron X-ray diffraction experiments. S.K., L.D., and F.I.A. processed the synchrotron X-ray diffraction data. Al.A., B.W., and M.Be. performed the theoretical calculations. L.D., N.D., S.K., M.By., and U.R. contextualized the data interpretation. L.D., N.D., and S.K. prepared the first draft of the paper with contributions from all other authors. All the authors commented on successive drafts and have given approval to the final version of the paper.

## Funding

## Competing interests

The authors declare no competing interests.
