## [Peer Review File · Nature Communications]

Extending carbon chemistry at high-pressure by synthesis of CaC_2 and Ca_3C_7 with deprotonated polyacene- and *para*-poly(indenoindene)-like nanoribbonsREVIEWER COMMENTS

Reviewer #2 (Remarks to the Author):

Dear Editor,

I have carefully read the manuscript "Extending carbon chemistry at high-pressure by synthesis of CaC_2 and Ca_3C_7 ..." by S. Khandarkhaeva and co-workers. The study focuses on the experimental synthesis of two novel calcium carbides: a high-pressure phase for the CaC_2 stoichiometry (HP- CaC_2) and a compound with new a stoichiometry, Ca_3C_7 . Very remarkably, in these carbides the anion moieties consist of ionic, pure carbon polymers made of fused six-membered rings (HP- CaC_2) and fused distorted six- and five-membered rings (Ca_3C_7). These polymeric forms were never found in carbides previously. The work represents a significant advance with respect to the study of reference [16], published on Nat. Comm., where the high-pressure formation of pure carbon polymeric anions was indeed DFT predicted but not experimentally observed. The conclusions of the present study are supported by a robust synchrotron, single crystal XRD analysis on samples in diamond anvils cells, treated by laser heating, together with DFT predictions which add to the experimental investigation by providing insights into the electronic charge distribution in the novel carbides. All the procedures are well described for the work to be reproduced. The study is of great importance for condensed matter chemistry, and materials and planetary science. I could recommend the publication of the manuscript on Nat. Comm., once the authors have addressed certain flaws that the study suffers from. Here I list my main comments:

- 1) the authors do not report Raman spectra for the new carbides. Is there any particular reason for that? Indeed, Raman peaks from the polymeric anions, in principle, would be highly informative and capable to provide physical insights into the nature of this polymers.
- 2) I could find neither pictures nor general visual description of the new carbides, after laser heating. Please, comment on this and maybe add pictures. This is particularly important, since one novel carbide is DFT predicted to be metallic, while the other one should be semiconductor.
- 3) I presume that the additional carbon content leading to the increased C/Ca stoichiometric ratio in Ca_3C_7 : 2.3 comes from the paraffine oil, isn't it? Please, comment on this. Also, is there any other way the oil may have contributed to the reported reactions? It would be desirable to check the results by using other, likely more inert pressure transmitting media.
- 4) The reported C-C distances between distinct nearest neighbour C pairs differ by at most a few percent in the new carbides. Is this result a sufficient demonstration of aromaticity for the carbon polymers? The authors should comment on this point and add a discussion on aromaticity in the manuscript.

Minor points:

- 1) I think the sentence: "On decompression at ambient temperature, featureless Raman spectra were observed up to ~ 1.8 GPa...", at page 7 should read: "On decompression at ambient temperature, featureless Raman spectra were observed down to ~ 1.8 GPa...".
- 2) In figure 3, left panel, the dashed line for the lattice parameter "a" should be red.
- 3) In the caption of figure S1, there is some inconsistency in the sentence "DilorXY system equipped with an Ar (excitation wavelength: 532 nm) laser source". Indeed, 532 nm should rather be the wavelength of a doubled Nd;YAG laser lasing ad 1064 nm.
- 4) Figure S8, the label for the horizontal axis is incorrect. It should be "Raman shift".

Reviewer #3 (Remarks to the Author):

The MS "Extending carbon chemistry at high-pressure by synthesis of CaC₂ and Ca₃C₇ with deprotonated polyacene- and para-poly(indenoindene)-like nanoribbons" is primarily an experimental work detailing the structural solution, refinement, and characterization of two calcium carbides synthesized at very high pressure and temperature conditions. The authors use complementary experimental techniques (viz., spectroscopy) and computational methods to support their single-crystal diffraction efforts. The work has potential significant impact in a number of respects: The structural motifs are interesting in their own right and thus of significant interest to the broad scientific community. It is a fine example of in situ high pressure single crystal diffraction (a technique that seems to not yet enjoy widespread use in the high pressure community). It constitutes an experimental confirmation of a previously predicted polymorph of CaC₂. It details possible structural motifs that could be found at simultaneous pressure and temperature conditions extant in deep planetary interiors. Finally, the authors suggest hydrolysis within planetary interiors of carbides such as those described in this work may be a source of aromatic compound(s).

My primary concern/criticism is a lack of details regarding the HP/HT synthesis of the metal carbides. The authors should combine qualitative, quantitative, and graphical presentations to provide a much more detailed account of the synthesis process and the resulting synthesized product. Specifically:

1. Graphically, it would be helpful to include a representative plot of the temperature measurement for one of the reported temperatures in Table S1. From that, a reader could immediately get information including: the wavelength range of the measurement, the general quality of the transfer function (demonstrated by the overall agreement between the corrected spectrum and the graybody fit), the approximate intensity of the measured signal (demonstrated by the amount of noise in the corrected spectrum), and the ease of heating control (demonstrated, for example, by relatively close agreement in the measured temperature on each side of the sample).
2. Quantitatively, it would be helpful to give some general parameters of the heating. Some details that could accompany a representative plot described above could include: the (peak) power of one or both lasers, the typical duration of the heating, and the number of times a sample was heated for synthesis at a given pressure (if more than once).
3. Qualitatively, it would be helpful to provide more information about the heating process and the resulting product. Did the sample readily couple with the laser? Was the sample stationary during heating? Or several spots/large areas were heated? This could be very important as the reported sample dimensions are on the order of tens of microns whereas the reported laser spot size is on the order of microns. At any one moment during the heating process there must be a very large radial temperature gradient with respect to the central heating spot. Related to this, was the synthesized sample dominated by the relevant carbide? Or there were just a few spots of the relevant carbide within the overall sample volume? The authors report that the sample was systematically explored (2D X-ray mapping) to find the best spot(s). Were these spots at or near the heating spot(s)? It is not clear if there were a wealth of spots to choose from, or rather, the authors were only able to find one or two spots within the sample volume.

I recognize that some of the suggested additions may not be available after the fact (e.g., perhaps the laser power was not recorded during heating) and this is okay. But the authors should add some of the suggested information (as well as some things I am likely missing) to provide a more complete description of the synthesis. These additions would belong almost entirely in the Supplementary Information, although some qualitative description could go in the main body. Note that the need for greater detail regarding the synthesis process is particularly important in this case, as the authors have reported interesting new metal carbides, but they have not reported a systematic study of, for example, the stability field(s) or phase diagram(s). This is fine, but then there needs to be enough information that others can follow with a more systematic exploration of the phase space related to various polymorphs or stoichiometries.

Some very minor comments:

The method(s) for pressure measurement should be stated (and the fitting/calibration scale for the method(s) should be referenced).

The Methods section describes Raman measurements using a LabRam system, but then the only measurements discussed (in the main text) and presented (in the SI) appear to be executed using a DilorXY system. Excitation wavelength of the latter is described as Ar at 532 nm; should this be 514 nm? Or is the laser a solid state YAG laser? In short, double-check the report of Raman measurements and the associated details.

In summary, the MS is an outstanding example of the structural solution, refinement, and characterization of two metal carbides synthesized at very high pressure and temperature conditions. The reported results are of general interest to the broad scientific community and have the potential to make a significant impact in related subfields. Its potential future impact could be significantly enhanced by adding more details regarding the synthesis.

Reviewer #4 (Remarks to the Author):

Khandarhaeva et al. present a noteworthy contribution reporting the synthesis of molecules consisting of fused C5 and C6 units from CaC₂, under high-pressure and high-temperature conditions. The reaction products and supported by density functional calculations are novel and deserve publication.

However, certain inconsistencies in the description of the structures merit clarification.

Firstly, the products are composed of fused C5-C6 and C6 rings. These molecules are not unlike anthracene, which is chemically a molecule, not a nanoribbon.

Secondly, there seems to be an overemphasis on the deprotonated para-poly(indenoindene) in describing the structure of Ca₃C₇. Given that the molecule originated from C₂ units, the absence of protons in the structure is no surprise.

From a chemical standpoint, the formation of a high-pressure CaC₂ structure at 41 GPa with fused C6 rings is anticipated. It is expected the C₂ units will combine, forming rings with an even number of atoms. Remarkably, at a slightly lower pressure of 38 GPa, a molecule featuring two naphthalene-like C6 units linked via a C5 ring is unusual. The formation of these molecules suggests the dissociation of C₂ units into atomic C before connecting the two naphthalene-like molecules. This observation indicates a deeper and more intricate chemistry underlying this process.

The pivotal question revolves around the novelty of this discovery and whether it warrants publication in Nature Communications. This paper lacks a detailed discussion of the potential implications of the novel high-pressure carbides and the underlying chemistry.

From the observation of Raman features relating to C-H vibrations in a sample prepared from compression and laser-heated CaC₂-I and Ca(OH)₂ followed by decompression, it was then postulated that hydrolysis of the carbides may be a source of poly-aromatic hydrocarbons (PAH). This proposition is mere speculation. There is no evidence to support the purported PAHs were produced from the high-pressure polycarbides. In fact, it is most likely originated from distinctively different chemical reactions.

While theoretical calculations successfully replicated the experimental structures, they failed to provide meaningful insights into the mechanisms responsible for forming these novel structures. A more in-depth exploration into the chemical processes governing the genesis of these unique carbides is a worthwhile pursuit.

The paper is not acceptable in its current form.

Please note, we have no "Reviewer #1"

Reviewer #2

I have carefully read the manuscript "Extending carbon chemistry at high-pressure by synthesis of CaC₂ and Ca₃C₇ ..." by S. Khandarkhaeva and co-workers. The study focuses on the experimental synthesis of two novel calcium carbides: a high-pressure phase for the CaC₂ stoichiometry (HP-CaC₂) and a compound with new a stoichiometry, Ca₃C₇. Very remarkably, in these carbides the anion moieties consist of ionic, pure carbon polymers made of fused six-membered rings (HP-CaC₂) and fused distorted six- and five-membered rings (Ca₃C₇). These polymeric forms were never found in carbides previously. The work represents a significant advance with respect to the study of reference [16], published on Nat. Comm., where the high-pressure formation of pure carbon polymeric anions was indeed DFT predicted but not experimentally observed. The conclusions of the present study are supported by a robust synchrotron, single crystal XRD analysis on samples in diamond anvils cells, treated by laser heating, together with DFT predictions which add to the experimental investigation by providing insights into the electronic charge distribution in the novel carbides. All the procedures are well described for the work to be reproduced. The study is of great importance for condensed matter chemistry, and materials and planetary science. I could recommend the publication of the manuscript on Nat. Comm., once the authors have addressed certain flaws that the study suffers from. Here I list my main comments:

We are very delighted by Reviewer's #2 assessment of our work and for his/her suggestions.

1) the authors do not report Raman spectra for the new carbides. Is there any particular reason for that? Indeed, Raman peaks from the polymeric anions, in principle, would be highly informative and capable to provide physical insights into the nature of this polymers.

As we mentioned in the manuscript, Raman spectra become featureless upon compression of CaC₂ over ~30 GPa (Fig. S1). Laser heating does not result in any changes in the Raman spectrum, as well as a decompression down to ~1.8 GPa (Fig. S8). A reason for such a behavior may be a metallic character of HP CaC₂ and very narrow-band semiconductor nature (on the border of metallicity) of Ca₃C₇ (even if metals may give a Raman signal, it is extremely weak).

2) I could find neither pictures nor general visual description of the new carbides, after laser heating. Please, comment on this and maybe add pictures. This is particularly important, since one novel carbide is DFT predicted to be metallic, while the other one should be semiconductor.

Upon Referee's #2 request, we have added photos of the samples after laser heating at ~38 GPa and ~93 GPa (Fig. S9). As expected for metallic and/or narrow-band semiconductor materials, the samples are dark.

3) I presume that the additional carbon content leading to the increased C/Ca stoichiometric ratio in Ca₃C₇: 2.3 comes from the paraffine oil, isn't it? Please, comment on this. Also, is there any other way the oil may have contributed to the reported reactions? It would be desirable to check the results by using other, likely more inert pressure transmitting media.

At this point, we cannot say for sure that additional carbon came from paraffin; diamond from the diamond anvils themselves can be a source of carbon, as we and many other groups reported before. Moreover, in runs #4, #5, and #6 pure CaC₂ was loaded without a PTM and at different pressures we observed similar products.

4) The reported C-C distances between distinct nearest neighbour C pairs differ by at most a few percent in the new carbides. Is this result a sufficient demonstration of aromaticity for the carbon polymers? The authors should comment on this point and add a discussion on aromaticity in the manuscript.

We fully agree with Referee #2 that the question about nature of the chemical bonding in new carbides is very interesting and important, and that is why we wouldn't like to jump with bold conclusions. Until now there are no systematic structural studies of the effect of pressure in the range of 50 to 100 GPa on bond lengths in different types of carbides, so we cannot make a firm conclusion if a small difference in the bond length is important or not. Moreover, experimental uncertainties in bond lengths make the discussion ambiguous, and in-depth theoretical analysis is beyond the scope of this work. So, we prefer to refrain from further discussions of chemical bonding in the new compounds.

Minor points:

1) I think the sentence: "On decompression at ambient temperature, featureless Raman spectra were observed up to ~1.8 GPa...", at page 7 should read: "On decompression at ambient temperature, featureless Raman spectra were observed down to ~1.8 GPa..."

Thanks, corrected.

2) In figure 3, left panel, the dashed line for the lattice parameter "a" should be red.

Thanks, corrected.

3) In the caption of figure S1, there is some inconsistency in the sentence “DilorXY system equipped with an Ar (excitation wavelength: 532 nm) laser source”. Indeed, 532 nm should rather be the wavelength of a doubled Nd:YAG laser lasing at 1064 nm.

Thanks, corrected.

4) Figure S8, the label for the horizontal axis is incorrect. It should be “Raman shift”.

Thank you, corrected.

Reviewer #3

The MS “Extending carbon chemistry at high-pressure by synthesis of CaC₂ and Ca₃C₇ with deprotonated polyacene- and para-poly(indenoindene)-like nanoribbons” is primarily an experimental work detailing the structural solution, refinement, and characterization of two calcium carbides synthesized at very high pressure and temperature conditions. The authors use complementary experimental techniques (viz., spectroscopy) and computational methods to support their single-crystal diffraction efforts. The work has potential significant impact in a number of respects: The structural motifs are interesting in their own right and thus of significant interest to the broad scientific community. It is a fine example of in situ high pressure single crystal diffraction (a technique that seems to not yet enjoy widespread use in the high pressure community). It constitutes an experimental confirmation of a previously predicted polymorph of CaC₂. It details possible structural motifs that could be found at simultaneous pressure and temperature conditions extant in deep planetary interiors. Finally, the authors suggest hydrolysis within planetary interiors of carbides such as those described in this work may be a source of aromatic compound(s).

We would like to thank Reviewer #3 for careful consideration of our work.

My primary concern/criticism is a lack of details regarding the HP/HT synthesis of the metal carbides. The authors should combine qualitative, quantitative, and graphical presentations to provide a much more detailed account of the synthesis process and the resulting synthesized product. Specifically:

- 1. Graphically, it would be helpful to include a representative plot of the temperature measurement for one of the reported temperatures in Table S1. From that, a reader could immediately get information including: the wavelength range of the measurement, the general quality of the transfer function (demonstrated by the overall agreement between the corrected spectrum and the graybody fit), the approximate intensity of the measured signal (demonstrated by the amount of noise in the corrected spectrum), and the ease of heating control (demonstrated, for example, by relatively close agreement in the measured temperature on each side of the sample).*

Following Reviewer's #3 request we have added Fig. S10 illustrating the quality of the Planck fit.

2. Quantitatively, it would be helpful to give some general parameters of the heating. Some details that could accompany a representative plot described above could include: the (peak) power of one or both lasers, the typical duration of the heating, and the number of times a sample was heated for synthesis at a given pressure (if more than once).

As a rule, a sample is easily coupled with the laser and its heating requires a relatively low power (~5 to ~15 W from each side depending on pressure). A heating duration varies from ~15 to ~60 seconds. This information has been added to the Methods. In the footnote of Table S1 we mention if a sample was heated not only at the home laboratory but also at a synchrotron facility.

3. Qualitatively, it would be helpful to provide more information about the heating process and the resulting product. Did the sample readily couple with the laser? Was the sample stationary during heating? Or several spots/large areas were heated? This could be very important as the reported sample dimensions are on the order of tens of microns whereas the reported laser spot size is on the order of microns. At any one moment during the heating process there must be a very large radial temperature gradient with respect to the central heating spot. Related to this, was the synthesized sample dominated by the relevant carbide? Or there were just a few spots of the relevant carbide within the overall sample volume? The authors report that the sample was systematically explored (2D X-ray mapping) to find the best spot(s). Were these spots at or near the heating spot(s)? It is not clear if there were a wealth of spots to choose from, or rather, the authors were only able to find one or two spots within the sample volume.

Two or three spots were heated at each sample and transformed materials were found in all heated spots. All laser-heating facility we employed are equipped with pi-shapers that provides flat-top distribution of laser energy and minimizes radial gradients. The X-ray mapping of the samples we used rather to find the best quality crystallites for further single crystal data collection than to check if carbides form inside or outside of the heating spot. Nevertheless, the maps show that the whole heated area contains novel carbides. All relevant information has been added to the Methods.

I recognize that some of the suggested additions may not be available after the fact (e.g., perhaps the laser power was not recorded during heating) and this is okay. But the authors should add some of the suggested information (as well as some things I am likely missing) to provide a more complete description of the synthesis. These additions would belong almost entirely in the Supplementary Information, although some qualitative description could go in the main body. Note that the need for greater detail regarding the synthesis process is particularly important in this case, as the authors have reported interesting new metal carbides, but they have not reported a systematic study of, for example, the stability field(s) or phase diagram(s). This is fine, but then there needs to be enough information that

others can follow with a more systematic exploration of the phase space related to various polymorphs or stoichiometries.

We did our best to respond to all Reviewer's #3 comments. However, as a matter of comment from our side, we are fully convinced that experimental details requested by Referee #3 concerning laser heating are excessive. For reproducing our experiments temperature to be achieved is important and it was specified in the original manuscript. The laser heating equipment and methodology of heating at the BGI, at many other laboratories all over the world, and at synchrotron facilities are state-of-the-art. The procedure of temperature measurements in laser-heated DACs is routine, and power required for laser heating strongly depends on a sample preparation and a type of optics used (particularly objectives).

Some very minor comments:

The method(s) for pressure measurement should be stated (and the fitting/calibration scale for the method(s) should be referenced).

The Methods section describes Raman measurements using a LabRam system, but then the only measurements discussed (in the main text) and presented (in the SI) appear to be executed using a DilorXY system. Excitation wavelength of the latter is described as Ar at 532 nm; should this be 514 nm? Or is the laser a solid state YAG laser? In short, double-check the report of Raman measurements and the associated details.

Thank you, corrected.

In summary, the MS is an outstanding example of the structural solution, refinement, and characterization of two metal carbides synthesized at very high pressure and temperature conditions. The reported results are of general interest to the broad scientific community and have the potential to make a significant impact in related subfields. Its potential future impact could be significantly enhanced by adding more details regarding the synthesis.

We thank Reviewer #3 for positive evaluation of our work.

Reviewer #4

Khandarhaeva et al. present a noteworthy contribution reporting the synthesis of molecules consisting of fused C5 and C6 units from CaC2, under high-pressure and high-temperature conditions. The reaction products and supported by density functional calculations are novel and deserve publication.

However, certain inconsistencies in the description of the structures merit clarification.

We do not see any inconsistencies in the description of the structures, but we are happy to provide Reviewer #4 all clarifications.

Firstly, the products are composed of fused C5-C6 and C6 rings. These molecules are not unlike anthracene, which is chemically a molecule, not a nanoribbon.

This statement is not quite clear for us; however, if we would follow Reviewer's #4 reasoning then ethylene would be the same as polyethylene, but for the syntheses of the latter the Nobel Prize was awarded. Note that "fused C5-C6 and C6 rings" are not "molecules" by any means.

Secondly, there seems to be an overemphasis on the deprotonated para-poly(indenoindene) in describing the structure of Ca₃C₇. Given that the molecule originated from C₂ units, the absence of protons in the structure is no surprise.

There is no "overemphasis" or "underemphasize" in the use of the word "deprotonated". This is just a term used in chemical nomenclature.

From a chemical standpoint, the formation of a high-pressure CaC₂ structure at 41 GPa with fused C₆ rings is anticipated. It is expected the C₂ units will combine, forming rings with an even number of atoms. Remarkably, at a slightly lower pressure of 38 GPa, a molecule featuring two naphthalene-like C₆ units linked via a C₅ ring is unusual. The formation of these molecules suggests the dissociation of C₂ units into atomic C before connecting the two naphthalene-like molecules. This observation indicates a deeper and more intricate chemistry underlying this process.

We agree with this statement of Reviewer #4.

The pivotal question revolves around the novelty of this discovery and whether it warrants publication in Nature Communications. This paper lacks a detailed discussion of the potential implications of the novel high-pressure carbides and the underlying chemistry.

In our paper, for the first time the predicted *Immm*-CaC₂ high-pressure polymorph of CaC₂ (HP-CaC₂) containing deprotonated polyacene-like carbon polyanion is reported. A novel Ca₃C₇ compound, never anticipated or observed before, with an orthorhombic crystal structure (space group *Pnma*) featuring infinite, fully deprotonated *para*-poly-indenoindene (*p*-PInIn)-like chains was discovered. Structures of both carbides were solved and refined using single crystal XRD at several pressures. For the first time we demonstrate that a compound with fused six-member carbon rings (reminiscent in shape of benzene) can be synthesised above 100 GPa. We also demonstrate

that on decompression in the presence of water HP-CaC₂ forms polycyclic aromatic hydrocarbons (PAHs). This observation may provide a possible abiotic route to polycyclic aromatic hydrocarbons in the Universe. We have no doubt in the novelty and importance of our work based on solid experimental data and supported by theoretical analysis. We do not see any need in speculative discussions of potential implications and underlying chemistry.

From the observation of Raman features relating to C-H vibrations in a sample prepared from compression and laser-heated CaC₂-I and Ca(OH)₂ followed by decompression, it was then postulated that hydrolysis of the carbides may be a source of poly-aromatic hydrocarbons (PAH). This proposition is mere speculation. There is no evidence to support the purported PAHs were produced from the high-pressure polycarbides. In fact, it is most likely originated from distinctively different chemical reactions.

Hydrolysis of carbides is a well-known phenomenon. Our experiment was designed to observe hydrolysis of novel carbides, and we got clear evidences that it happened. If Reviewer #4 would have provided any alternative explanation of our observations, we could consider/discuss it, but Reviewer' #4 statement above does not provide any matter to be debated.

While theoretical calculations successfully replicated the experimental structures, they failed to provide meaningful insights into the mechanisms responsible for forming these novel structures. A more in-depth exploration into the chemical processes governing the genesis of these unique carbides is a worthwhile pursuit.

We highly appreciate Reviewer's #4 suggestion. As known, search for "mechanisms responsible for forming novel structures" often requires years of efforts of large groups of scientists. We certainly will work in this direction and will publish our findings (if any).

REVIEWERS' COMMENTS

Reviewer #2 (Remarks to the Author):

The authors' reply to all my criticisms is convincing, and I think the authors also replied properly to all referee's #3 and #4 concerns. I can recommend the publication of the revised manuscript on Nature Communications as is.

Reviewer #3 (Remarks to the Author):

Khandarkhaeva and coauthors have submitted a revised manuscript "Extending carbon chemistry at high-pressure by synthesis of CaC₂ and Ca₃C₇ with 2 deprotonated polyacene- and para-poly(indenoindene)-like nanoribbons."

I had a very favorable impression of the original manuscript. My primary reservation was what I considered to be a lack of sufficient detail regarding the synthesis of the novel polymorph and carbide. The authors have made acceptable revisions to better describe the synthesis process. I raised some minor points which were also properly addressed in the revised manuscript.

While there could very well be additional, related research on these compounds in the future (stability fields, transition mechanisms, etc.), this work can stand on its own. I here echo my original review that the work can have a significant impact, not only in the extreme conditions community, but also in the broader scientific community. I believe it is suitable for publication in Nature Communications in its current form.

Reviewer #4 (Remarks to the Author):

The responses from the authors have helped to settle my queries on the crystal structures. The report of the first observation of polymeric structures of fused aromatic carbon rings at high pressure merits publication. Therefore, the contribution is acceptable.

However, I am still somewhat disappointed that the theoretical results were only used to justify the observation and have not been fully utilized to propose possible mechanisms on the formation of these novel structures.